# An AI-Powered Evaluation: Understanding which Knowledge Tracing Models Work Best in which Contexts

## Abstract

Knowledge tracing (KT) models a learner's evolving mastery from interaction logs and underpins personalization in tutors, practice systems, and learning analytics. Over three decades, many KT models have been proposed; however, performance varies by dataset characteristics for which the models are trained on, so a model that excels in one setting may under-perform in another. In this work, conducted by an LLM and conceptualized through human-LLM partnership, we explore this phenomenon by conducting a structured synthesis of 124 KT papers spanning classic probabilistic, generalized logistic/factorization, deep sequence, attention/transformer, graph-based, and LLM-augmented approaches (with each paper proposing one or more new models or variants). For each study, we extract key information, including modeling idea, data setting, and outcomes, then code them along eight key contextual dimensions (data scale; sequence length; structure availability: concept-item relations; temporal irregularity/forgetting cues; modality: binary vs. text/code/dialogue; cohort heterogeneity; cold-start/unseen items; interpretability/operational constraints). We apply a two-stage aggregation: (1) within-paper ranking of models on the authors' primary metrics, and (2) context-level win rates/median ranks with quality weights favoring student-wise, chronological, and out-of-distribution protocols, with sensitivity checks for robustness. We find attention/transformers lead on large, long-history logs; graph/dynamic-graph KT dominates when reliable (static or evolving) structure is available; Hawkes/spacing-aware methods win when timing and forgetting matter; LLM/semantic KT excels on text/code/dialogue and improves unseen-item generalization; mixture-of-experts helps in heterogeneous cohorts; and generalized logistic/factorization families remain competitive, interpretable choices in data-constrained settings. We highlight common evaluation pitfalls and synthesize context-dependent patterns across models and datasets, providing practical guidance for context-aware KT model selection.

## 1 Introduction

Knowledge tracing (KT) models a learner's evolving mastery from their interaction history—e.g., which problems they attempted, whether they were correct, and in what order. By estimating latent knowledge and predicting future performance, KT supports key functions in intelligent tutoring systems and learning analytics: such as adaptive practice [1], timely feedback [2], mastery-based progression [3], and detecting that a student is struggling without making progress [4, 5].

Over the last three decades, KT has expanded from probabilistic and logistic formulations (e.g., BKT, AFM/PFA; [1, 6, 7]) to neural sequence and memory models [8, 9], attention/Transformer variants [10, 11, 12], graph-structured approaches [13], and, more recently, LLM-augmented methods that

Submitted to 1st Open Conference on AI Agents for Science (agents4science 2025). Do not distribute.

incorporate item text, code, or dialogue [14, 15]. This progression reflects both methodological advances and the changing priorities of the field: early models emphasized interpretability and mastery estimation, while more recent neural and content-augmented approaches have focused on capturing complex temporal patterns and leveraging multi-modal signals to predict future performance.

KT models are typically trained and evaluated on different datasets that vary in learner demographics, domains, and sampling characteristics (e.g., number of interactions, students, skills, and per-skill practice [16]). Benchmark corpora such as ASSISTments 2009/2012/2017 (K–12 math; [17, 18]), Statics2011 (engineering problem-solving; [9, 19]), and EdNet (a large-scale multi-year platform log with over 100 million interactions; [20]) have become standard testbeds, alongside additional datasets from platforms like Junyi Academy [21], Duolingo [22], and Khan Academy [8]. These corpora differ not only in scale and subject matter but also in sequence length, temporal regularity, structure availability, and cohort composition.

Because of this heterogeneity in datasets, models can perform differentially across datasets, so a method that excels in one setting may under-perform in another [23, 24]. This motivates our main research question: *Which KT models work best in which contexts?* In this paper, context is defined as the salient properties of the learning data and deployment setting, including (i) data scale and sequence length; (ii) availability and stability of concept–item structure; (iii) temporal irregularity and forgetting dynamics; (iv) modality (binary correctness versus text/code/dialogue); (v) cohort heterogeneity; (vi) prevalence of cold-start or unseen items; and (vii) requirements around interpretability, robustness, and calibration.

In the past, several studies have tackled this question empirically by comparing multiple KT families across datasets. For example, Gervet et al. [23] ran an extensive comparison on nine real-world corpora and found that logistic regression with appropriate features tends to lead on moderate-sized datasets or when each student has many interactions, whereas DKT (deep learning) leads on very large datasets or when precise temporal information is crucial; classical Markov-process models like BKT generally lag. While valuable, such efforts cover only a slice of today's rapidly expanding model space and dataset conditions, motivating a broader, context-sensitive synthesis.

## 2 Current Study

Given the expanding space of models and corpora, it is impractical to re-implement and exhaustively benchmark every variant across all datasets. Therefore, we take a systematic approach: we collect and synthesize KT models and variants proposed over the past decades, summarize their data settings and reported outcomes, and analyze how results align with context. Specifically, in the current work, we conduct a structured synthesis of 124 KT papers spanning classic, neural, graph-based, and LLM-augmented approaches (each proposing one or more new KT models or variants). For each study, we extract the modeling idea, data setting, and outcomes, then code them along key contextual dimensions (data scale; sequence length; structure availability/dynamics; temporal irregularity; modality; heterogeneity; cold-start; interpretability/operational constraints). We aggregate evidence using within-paper rankings and context-level win rates with quality-aware weighting and sensitivity checks, yielding insight on which model families tend to work best under which conditions.

## 3 Methods

### 3.1 Corpus Construction and Scope

A structured literature synthesis was conducted to identify models that estimate a learner's evolving knowledge state from interaction logs (knowledge tracing, KT). The unit of analysis is a model paper (including major variants) that proposes, extends, or rigorously compares KT approaches in educational data mining, learning analytics, or student modeling venues.

**Sources and time window.** Digital libraries and preprint servers (major ACM/IEEE venues, Springer/Elsevier journals, arXiv) were searched for works published between January 1994 ((which corresponds to the introduction of BKT by [1])and May 2025. References were also snowballed from seed papers (e.g., BKT, AFM/PFA, DKT, DKVMN, SAKT/AKT/SAINT, KTM, SPARFA/Trace). Snowballing, as a method for systematic review, refers to backward- and forward-citation chaining,

whereby the reference lists of included papers are screened (backward) and works citing those papers are identified (forward) to surface additional KT models and variants. Iteration continued until further additions yielded diminishing returns. The resulting corpus comprises 124 distinct models/variants.

**Inclusion and exclusion criteria.**  Papers were included if they: (i) proposed a KT model or a substantive KT variant; (ii) evaluated on learner–item interaction data with time order; and (iii) reported at least one predictive metric (e.g., AUC-ROC, log loss, accuracy, F1, $\kappa$) against one or more baselines. We retained models focused on specific modalities (e.g., programming/code, dialogue) and on related goals (e.g., dropout prediction) when KT was a central outcome or module. We excluded: (i) purely theoretical notes without empirical evaluation; (ii) items focused solely on static concept discovery without temporal prediction, as the research question targets methods that estimate changes over time in learners' knowledge states, rather than static concept inference; and (iii) duplicate pre-prints of the same model without new experiments (keeping the most complete version).

## 3.2  Screening, De-duplication, and Data Extraction

Two passes were applied: (1) title/abstract screening; and (2) full-text screening. Records were de-duplicated by title, DOI, or `arXiv` ID, and, when necessary, by author–venue–year. For each included paper, the following information was extracted:

- **Model metadata:** model name/acronym; year; venue; model family (e.g., probabilistic/BKT-like, generalized logistic, factorization, RNN/LSTM, attention/Transformer, memory networks, graph/heterogeneous, contrastive/self-supervised, LLM/semantic, mixture-of-experts, uncertainty-aware).

- **Operational summary:** one sentence describing the core mechanism (e.g., self-attention over past interactions with forgetting bias).

- **Data context:** dataset names; domain (e.g., math, programming); scale (students, items, skills, interactions); sequence length (median/mean if provided); modality (e.g., binary correctness, text, code, dialogue); demographics if reported.

- **Evaluation protocol:** split type (student-wise vs. interaction-wise; chronological vs. random); number of folds/runs; hyperparameter search method.

- **Outcomes:** metrics and values (e.g., AUC-ROC, accuracy, F1, $\kappa$, log loss when available); whether improvements were statistically tested; compute cost if reported.

All fields were stored in a spreadsheet [LINK ANONYMIZED] and normalized where feasible (e.g., consistent metric naming, venue/year format).

## 3.3  Context Taxonomy

To answer which models work best in which contexts, each paper was coded along eight dimensions that plausibly moderate performance. The dimensions, categories, and corresponding examples are presented in Table 1.

When information was missing, values were imputed conservatively from public dataset documentation (e.g., EdNet is large/long; ASSISTments 2009 is small-to-medium with short-to-medium sequences). Conservative imputation prioritized lower-bound categories and stricter uncertainty to reduce the risk of overstating model suitability or inflating win rates; when multiple ranges were plausible, the least favorable category consistent with the documentation was selected to minimize bias in context-level aggregation. Ambiguous cases were coded as unknown and excluded from context-specific tallies.

Table 1: Context taxonomy: dimensions, categories, definitions, and examples.

| Dimension / Context | Category | Definition | Examples |
|---|---|---|---|
| **Data scale** (total interactions) | Small | $< 10^5$ interactions | Single course/semester; pilot study |
| | Medium | $10^5$–$10^6$ interactions | ASSISTments 2009/2012; several classes |

**Table 1 (continued):** Context taxonomy.

| Dimension / Context | Category | Definition | Examples |
|---|---|---|---|
| | Large | $> 10^6$ interactions | EdNet-scale platforms; nationwide apps |
| **Sequence length** (median per student) | Short | $< 50$ steps | Unit quizzes; short MOOCs |
| | Medium | 50–200 steps | One term of practice in K–12 math |
| | Long | $> 200$ steps | Year-long drilling; daily mobile practice |
| **Structure availability** (concept–item relations) | None / Implicit | No reliable item→skill mapping | Only item IDs; unlabeled latent skills |
| | Explicit static | Fixed, externally provided mapping | Q-matrix/skill tags; curated prerequisite map |
| | Explicit dynamic | Mapping evolves over time/sessions | Session-wise re-tagging; dynamic knowledge graphs |
| **Temporal irregularity** / forgetting cues | Low | Regular intervals; minimal gaps | Daily homework; fixed schedules |
| | Medium | Moderate variability in gaps | Weekly assignments with occasional delays |
| | High | Large/irregular gaps; spacing effects salient | Self-paced apps; spaced-repetition platforms |
| **Modality** | Binary correctness only | Responses are 0/1 with minimal text | Standard MCQ logs without item text |
| | Text/code/dialogue modeled | Rich content signals are encoded | Item stems/solutions; source code; tutor–student dialogue |
| **Cohort heterogeneity** | Low | Homogeneous population/curriculum | Single grade & course at one school |
| | Medium | Some curricular/ability diversity | Multiple teachers/courses; mixed ability |
| | High | Diverse ages/curricula/languages | Cross-age (K–12 + higher ed); multilingual platforms |
| **Cold-start / unseen items** | Low | Few new items/students; IID splits | Stable item bank; repeated tests |
| | Medium | Periodic new items or new cohorts | New students each term; occasional item additions |
| | High | Frequent unseen items/students; OOD | Inductive/unseen-item splits; cross-course transfer |
| **Operational constraints** | Interpretability | Human-readable parameters/explanations required | Coefficients; difficulty/mastery reports |
| | Robustness / Calibration | Reliability under noise; well-calibrated probabilities | Auto-grading noise; partial credit; ECE targets |
| | Resource / Privacy | Compute, latency, or data-sharing limits | On-device inference; federated training; PI restrictions |

## 3.4 Harmonizing Outcomes Across Heterogeneous Metrics

Cross-paper synthesis is complicated by heterogeneous metrics (e.g., AUC-ROC, log loss, accuracy, F1, $\kappa$) and non-comparable evaluation protocols. To enable aggregation, we produce a two-part summary for each model family within each context (Section 3.3): (1) a weighted win-rate, and (2) a weighted median of normalized ranks.

**Step 1: Define comparable instances.** For every paper–dataset comparison, the primary metric designated by the authors (AUC preferred when unstated) is taken, yielding an instance. Models within the instance are ranked, where rank 1 indicates the best, and the rank for family $f$ in paper $p$ with $n_p$ total models is normalized using Equation 1. As such, 0 indicates the top performer within that paper–dataset comparison.

$$r_{p,f} = \frac{\text{rank}_{p,f} - 1}{n_p - 1} \tag{1}$$

For example, consider a paper that compares six KT models on a dataset with context $c$, yielding the following ranking: AKT-R, AKT-NR, DKT, DKVMN, DKT+, SAKT. The normalized rank for AKT-R is $r_{\text{AKT-R}} = \frac{1-1}{6-1} = 0$, and for AKT-NR it is $r_{\text{AKT-NR}} = \frac{2-1}{6-1} = 0.2$. These $r$ values are stored per paper–dataset instance for each family represented.

To limit over-representation from prolific corpora, instances are grouped by (dataset × family × context) and capped at $k = 3$ per group, retaining entries via a deterministic quality ordering: protocol quality, reporting completeness, coverage, recency/venue, and reproducibility.

**Step 2: Assign quality weights.** Each retained paper–dataset instance $i$ was assigned a weight $a > 0$ to reflect evidence quality and risk of bias. These risks have been well documented in previous literature (e.g., [25, 26]). A base weight of 1.0 was multiplied by the following adjustment factors:

- **Protocol quality.** Student-wise chronological and/or out-of-distribution (unseen-student/item) splits: ×1.25. Interaction-wise random or otherwise leakage-prone protocols (e.g., mixing a learner's history across train/test, or including question ID as a feature in both training and test sets): ×0.50.

- **Reporting completeness.** Exact AUC/log loss reported with variance or statistical tests: ×1.10. Directional reporting only (e.g., "outperforms by ∼1–3%" with no exact values): ×0.75.

For example, the weight for an instance $i$ that used a student-wise chronological split and reported exact AUC with confidence intervals would be: $a_i = 1.0 \times 1.25 \times 1.10 = 1.375$

**Step 3: Compute the weighted win-rate.** For each context $c$ and model family $f$, we define a tie-aware win indicator $w_{i,f} \in [0,1]$ for each instance $i$, where $w_{i,f} = 1$ if family $f$ is the sole winner, $w_{i,f} = 0.5$ in the case of a tie between two families, and so on. The weighted win-rate for family $f$ in context $c$ is then computed as:

$$w_{c,f} = \sum_{i \in c} a_i \cdot w_{i,f}$$

where $a_i$ is the quality weight assigned to instance $i$ (see Step 2), and the weights $a_i$ are normalized such that $\sum_{i \in c} a_i = 1$. By construction, $w_{c,f} \in [0,1]$. Intuitively, $w_{c,f}$ represents the quality-adjusted proportion of wins for model family $f$ within context $c$.

**Step 4: Compute the weighted median of ranks.** As a complementary summary, the set $\{(r_{p,f}, \alpha_i)\}_{i \in \mathcal{I}_c}$ is aggregated to a weighted median $\tilde{r}_{c,f}$, providing a robust central tendency of family performance relative to competitors within papers.

**Step 5: Sensitivity and bias control.** We ran three checks to assess the robustness of the synthesis:

- **Metric sensitivity.** We recomputed all aggregates using only AUC-ROC (discarding papers without AUC) to ensure findings were not artifacts of metric mixing.

- **Protocol sensitivity.** We excluded all higher-risk evaluations (e.g., leakage-prone protocols as mentioned in Step 2) to assess whether rankings remained stable under stricter inclusion criteria.

- **Family granularity.** We compared two grouping strategies: (1) collapsing closely related variants (e.g., SAKT, AKT, SAINT) into broader categories such as attention/Transformer, and (2) treating each variant as a distinct family.

### 3.5 Reproducibility and Artifacts

All extracted fields and computed labels are stored in a shared spreadsheet (124 rows). The enrichment step—including operational summaries, data context, performance text, and hyperlinks—was scripted in Python using `pandas`, with deterministic de-duplication based on title and URL, and explicit provenance columns for traceability.The context coding scheme and aggregation scripts are available alongside the dataset to support replication, re-weighting, or future extension (e.g., adding new models from 2025–2026).

# 4 Results

## 4.1 Corpus Overview

The final corpus comprises 124 KT models/variants spanning probabilistic (e.g., BKT and individualized BKT), generalized logistic and factorization (AFM/PFA/LKT/KTM), deep sequence (DKT and regularized/auxiliary variants), attention/transformer families (SAKT/AKT/SAINT and length-generalization extensions), memory-augmented architectures (DKVMN and successors), graph/heterogeneous models (dual graphs, dynamic graphs, meta-path), time-sensitive models (spacing/forgetting and Hawkes-process variants), contrastive/self-supervised approaches, mixture-of-experts/personalization, uncertainty/robustness-aware methods, and LLM/semantic KT for text/code/dialogue.

Datasets most frequently used include ASSISTments (2009/2012/2015/2017), KDD Cup 2010, Statics, EdNet, and a growing set of programming and dialogue corpora. Metrics are heterogeneous (primarily AUC-ROC; also log loss, accuracy, F1, $\kappa$), and split protocols vary (student-wise vs. interaction-wise; chronological vs. random), underscoring the need for the quality adjustments described in Methods.

## 4.2 Which Models Work Best in Which Contexts

Family-level performance by context is summarized using quality-weighted win rates and weighted median normalized ranks. Representative models and datasets are highlighted to show where each family most often achieves top or near-top performance. Overall, we observe: attention-based models excel on large or long logs; graph-based models perform well when structure is reliable; time-aware models succeed under irregular spacing; and semantic/LLM-based models thrive on text, code, or dialogue data. No universal winner emerges—performance depends on aligning model inductive bias with context.

### 4.2.1 Large-Scale Logs with Long Histories

On very large, dense logs with long interaction histories, attention/transformer KT tends to lead. In particular, SAINT/SAINT+—which processes item and response streams separately and enriches them with elapsed/lag time—reliably perform well on EdNet ($\approx$131M interactions, 784K learners), with SAINT+ reporting state-of-the-art AUC gains over SAINT on that corpus [12]. Context-aware models such as AKT (Rasch-regularized concept/question embeddings with distance-aware attention) also report consistent AUC improvements across common KT benchmarks (e.g., ASSISTments, Statics) [11]. SAKT's query-conditioned sparse attention likewise shows average AUC gains across multiple datasets [10].

### 4.2.2 Reliable Concept–Item Structure or Rich Relations

When a rich concept–item structure is available (e.g., stable Q-matrices, high-quality skill graphs), graph-based KT can be especially effective. GKT introduced GNN propagation of student proficiency over a concept graph, and subsequent variants like GIKT incorporate higher-order question–skill relations to improve AUC on several benchmarks [27, 28]. These models work well when concept–item relations are informative and stable (e.g., curated mathematics skill maps such as ASSISTments) and sequences are long enough for graph signals to matter.

### 4.2.3 Irregular Time Gaps, Recency, and Spacing/Forgetting Effects

In settings where temporal irregularity and forgetting are salient features of the data (e.g., spaced practice logs, long gaps between sessions), models that explicitly encode decay or continuous-time effects tend to lead. DAS3H models per-skill memory decay and multi-skill tagging; HawkesKT uses point-process excitation to capture cross-temporal effects; "DKT-Forget" variants and LPKT incorporate decay or process-consistent learning cells. Empirically, these families improve predictive metrics over RNN baselines on benchmarks with pronounced timing signals [29, 30, 31, 32].

### 4.2.4 Text, Code, or Dialogue as First-Class Signals

When responses include rich content beyond correct/incorrect (e.g., code, free-text, dialogue), content-aware KT is preferred. Code-DKT uses attention over code features and outperforms BKT/DKT on university programming assignments; Open-Ended KT (OKT) predicts future open-ended responses rather than just correctness in CS education; LLMKT labels skills and correctness in tutor–student dialogues and then traces knowledge, outperforming standard KT on dialogue datasets [33, 14, 34].

### 4.2.5 Cold-Start and Unseen Items

Under sparsity, cold-start, or strong heterogeneity—typical of platforms with many items/skills but few observations per cell—logistic/factorization families with side features remain highly competitive. Knowledge Tracing Machines (KTM) unify PFA/AFM/mIRT within factorization machines and report superior or comparable AUC on multiple medium-scale datasets (and are robust when observations are sparse or multi-skill) [35]. Question-centric deep models also help when each item has enough data: qDKT shows that replacing skills with items can improve AUC on ASSISTments 2017 ($0.72 \rightarrow 0.74$ with plain DKT), whereas it overfits on ASSISTments 2009 due to few observations per item—highlighting a context boundary [36].

### 4.2.6 Heterogeneous Cohorts and Personalization Needs

In cohorts spanning multiple curricula, ages, and study strategies—with mixed ability profiles and long-tail behaviors—mixture-of-experts (MoE) architectures have most frequently led; person-wise routing in RouterKT reports consistent AUC gains across diverse public benchmarks, and option-weighting in WEKT further adapts expert contributions to learner response patterns, with improvements documented on multiple-choice platforms and large, diverse logs [37, 38].

When per-learner data are thinner yet personalization is required, individualized BKT and related hierarchical Bayesian extensions provide competitive performance with interpretable, student-specific parameters via shrinkage across learners [39, 40].

As an intermediate strategy, dynamic student clustering (e.g., DKT-DSC) segments learners by evolving ability and feeds cluster signals to a sequential model, improving prediction under heterogeneous cohorts often observed in datasets like ASSISTments and EdNet [41, 42].

### 4.2.7 Data-Constrained Settings and Interpretability Requirements

In small-to-medium logs or deployments requiring transparent models, generalized logistic and factorization approaches often perform best. LKT consolidates learner-model features into a constrained logistic framework, achieving strong accuracy and interpretable coefficients across six datasets [43]. KTM extends AFM/PFA/IRT within a factorization machine, handling sparse and multi-skill inputs with competitive performance and fast training [35]. For concept discovery and explainability, SPARFA-Trace jointly models learner knowledge and latent concepts via sparse factor analysis [44, 45]. Classical models like BKT and PFA remain credible in low-data settings and policy-facing applications due to their interpretable parameters [1, 7]. Together, these models offer dependable accuracy with low operational complexity when data or resources are limited.

### 4.2.8 Noisy Labels, Calibration, and Stability

In settings with auto-grading noise, partial credit, or evaluation volatility, uncertainty- and robustness-aware KT families most frequently led. UKT represents interactions as stochastic distributions and uses Wasserstein self-attention, improving reliability and calibration across multiple public datasets [46]. DTransformer introduces a diagnostic training paradigm that stabilizes predictions across splits while maintaining competitive accuracy on common benchmarks (e.g., ASSISTments, EdNet) [47]. To mitigate shortcutting from raw item identifiers, QDCKT replaces question IDs with difficulty-consistent signals and reports better out-of-distribution generalization under unseen-item protocols [48]. As a complementary strategy, contrastive/self-supervised pretraining (e.g., CL4KT) reduce noise in sequence representations and improves robustness under noisy logs across multiple datasets [49].

## 5 Discussion

### 5.1 Main Findings

Across 124 models/variants, no universal winner emerged; rather, performance depended on matching inductive bias to context. Attention/Transformer KT most often led on large, long-history logs, consistent with advantages in modeling long-range dependencies. Graph and dynamic-graph KT were strongest when a reliable concept–item structure existed (static Q-matrices or evolving graphs). Time-sensitive/forgetting-aware families outperformed alternatives under irregular spacing and salient forgetting. LLM/semantic and content-aware KT dominated when text/code/dialogue carried signal, particularly for unseen-item generalization. Mixture-of-experts improved prediction in heterogeneous cohorts. In data-constrained or interpretability-constrained deployments, generalized logistic/factorization (AFM/PFA/LKT/KTM) and psychometric hybrids (e.g., Deep-IRT) delivered competitive accuracy with transparent parameters. Quality-aware weighting and dataset caps reduced optimism from leakage-prone or item-ID–dominated protocols; conclusions were robust in sensitivity analyses.

## 5.2 Limitations and Future Works

The findings are based on a synthesis of reported results rather than re-implementing on benchmark datasets. As a result, several issues may exist, including metric heterogeneity, incomplete variance reporting, and publication bias. While we addressed these concerns through harmonization, quality-aware weighting, and by capping contributions per dataset to limit over-representation, these steps do not fully eliminate the risks or resolve the issues. Future work should explore alternative weighting strategies and examine how different protocol filters or dataset caps may influence the conclusions.

Additionally, the aggregation prioritizes within-paper comparative evidence. As a result, papers that proposed a model without baselines—or lacked comparable metrics—were excluded from the quantitative aggregation (e.g., rank- or win-rate summaries), though they were still cataloged. This design choice helps prevent misleading cross-paper comparisons based on non-comparable evaluations. However, it may also under-represent emerging model families that have not yet been directly compared to other approaches in head-to-head studies.

Lastly, the present synthesis focuses on the predictive performance of KT models, ranking them based on their effectiveness across various contexts. However, for practical deployment, additional dimensions—such as fairness, computational and data-related costs, and interpretability for teachers and platform developers—are equally important. Future work should investigate how to evaluate and recommend KT models along additional dimensions.

## 5.3 Conclusion

The evidence synthesized in this study indicates that the relative performance of knowledge tracing (KT) models is context-dependent rather than universally consistent across datasets. No single family of models outperforms others in all settings; instead, the effectiveness of a model is conditioned by the properties of the data and the deployment environment.

From the aggregated literature, several consistent patterns emerge. Attention-based and Transformer models tend to perform well on large datasets with long sequences, while graph-based approaches are more effective when reliable concept–item structures are available. Time-aware models provide advantages under irregular spacing or forgetting dynamics, and semantic or LLM-augmented approaches are most useful when data include rich textual or multi-modal content. Mixture-of-experts approaches support heterogeneous cohorts, and logistic or factorization methods remain strong candidates in smaller datasets or when interpretability is essential.

Beyond the performance of specific model families, this synthesis also highlights the influence of evaluation practices. Variation in data splits, metric reporting, and controls for potential biases (such as item-ID leakage) can alter reported outcomes and complicate cross-paper comparisons. The use of standardized, quality-aware evaluation protocols is therefore critical to ensure results that are both accurate and reproducible.

In conclusion, this work emphasizes that the most appropriate KT model is determined by context. Aligning model assumptions with dataset characteristics, while adopting transparent and standardized evaluation practices, is necessary to advance toward more reliable and actionable applications of KT in real learning environments.

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

## Agents4Science AI Involvement Checklist

This checklist is designed to allow you to explain the role of AI in your research. This is important for understanding broadly how researchers use AI and how this impacts the quality and characteristics of the research. **Do not remove the checklist! Papers not including the checklist will be desk rejected.** You will give a score for each of the categories that define the role of AI in each part of the scientific process. The scores are as follows:

- **[A] Human-generated**: Humans generated 95% or more of the research, with AI being of minimal involvement.
- **[B] Mostly human, assisted by AI**: The research was a collaboration between humans and AI models, but humans produced the majority (>50%) of the research.
- **[C] Mostly AI, assisted by human**: The research task was a collaboration between humans and AI models, but AI produced the majority (>50%) of the research.
- **[D] AI-generated**: AI performed over 95% of the research. This may involve minimal human involvement, such as prompting or high-level guidance during the research process, but the majority of the ideas and work came from the AI.

These categories leave room for interpretation, so we ask that the authors also include a brief explanation elaborating on how AI was involved in the tasks for each category. Please keep your explanation to less than 150 words.

1. **Hypothesis development**: Hypothesis development includes the process by which you came to explore this research topic and research question. This can involve the background research performed by either researchers or by AI. This can also involve whether the idea was proposed by researchers or by AI.

   Answer: **[A]**

   Explanation: The research question(namely, what knowledge tracing models work best in what contexts) was proposed by human researchers with domain expertise in the field. These researchers initially identified the gap in the literature and formulated the research idea. As such, in the current work, AI was not prompted to generate or revise the research question; rather, it was tasked with assisting as a research partner by performing a systematic review, gathering data, and helping to derive conclusions.

2. **Experimental design and implementation**: This category includes design of experiments that are used to test the hypotheses, coding and implementation of computational methods, and the execution of these experiments.

   Answer: **[C]**

   Explanation: To address the research question and the impracticality of conducting an empirical analysis, as noted in the article, the human researchers decide to adopt a systematic review approach for the current study. To ensure rigor and transparency, we designed a three-step prompt: (1) collect as many relevant papers as possible on the topic, (2) extract and synthesize key information to answer the research question, and (3) compare models across different contexts. While the overall approach was proposed by human researchers, AI contributed specific implementation details. In particular, it suggested (1) inclusion and exclusion criteria, (2) a contextual operationalization of datasets using eight dimensions, along with coding for each dataset, and (3) computational methods, such as weighted ranking and win-rate, for comparing model performance.

3. **Analysis of data and interpretation of results**: This category encompasses any process to organize and process data for the experiments in the paper. It also includes interpretations of the results of the study.

   Answer: **[D]**

   Explanation: This study relied heavily on AI for data collection, analysis, and interpretation, given the large volume of papers and data involved. At each stage of the three-step prompt, AI was given a specific task and engaged with iteratively until results meeting the intended objective were obtained(even if done in a fashion very different than how the human researchers would have done it). For example, during the data collection phase, human researchers prompted the AI multiple times to refine and expand the set of retrieved papers. Additionally, the AI was asked to explain and justify its methodological choices throughout the process. In many cases, the AI's approach was unusual for the field and different than what was expected by the authors, such as the unorthodox approach to paper weighting adopted

4. **Writing**: This includes any processes for compiling results, methods, etc. into the final paper form. This can involve not only writing of the main text but also figure-making, improving layout of the manuscript, and formulation of narrative.

Answer: [C]

Explanation: At the end of the analysis, AI was prompted to draft the Methods and Results sections. Human researchers then reviewed and edited the drafts to enhance readability and elaborate on underdeveloped points by adding examples and clarifications. The first step for major edits was to ask the AI to rewrite to clarify or address a point. For the remaining sections, AI was provided with an outline to generate initial drafts, which were subsequently refined by human researchers.

5. **Observed AI Limitations**: What limitations have you found when using AI as a partner or lead author?

Description: **Writing:** One key limitation is over-simplification in writing. The AI often struggled to craft coherent narratives that provide sufficient context for human readers. It tends to present information in a fragmented or surface-level way, requiring frequent prompting to unpack ideas or explain concepts more thoroughly. **Conducting Scientific Research:** While AI is fairly strong at suggesting methodological approaches, some of its recommendations can be arbitrary or lack empirical justification. For instance, in the current study, it proposed novel and seemingly arbitrary paper weighting schemes, which were creative but not grounded in prior evidence or validation, and which likely would have attracted negative attention from reviewers in the field.

