# OpenReview forum: "An AI-Powered Evaluation: Understanding which Knowledge Tracing Models Work Best in which Contexts"
_Agents4Science/2025/Conference — Submitted to Agents4Science_

### Official Review · Reviewer_LXP3 · 2025-10-03
**.**

**Clarity:** 3
**Significance:** 2
**Originality:** 4
**Overall:** 3
**Confidence:** 2

**Summary:**

This survey paper provides an analysis of Knowledge Tracing literature. This work is valuable in several ways: it demonstrates how AI can assist with large-scale literature synthesis, it covers 124 models/variants across multiple KT families, and it attempts to provide context-dependent guidance. The paper is well written and easy to read.

However I do believe that for a survey this is not systematic enough. To make a few examples:

* the paper lacks systematic search protocols. no structured queries, no reproducible screening methods. I understand the focus is automation, but in this context, the lack of systematic rigor does not allow me to assess whether the collection is complete or representative. Expressions like "Iteration continued until further additions yielded diminishing returns." - need to be more precise for understanding and reproducibility.

* each step of the pipeline seems to be done by the LLM, but to my understanding no validation is provided for the accuracy (e.g., taxonomy classifier). This is a bit concerning. without an additional eval, these downstream conclusions become unreliable.

In summary, this is a nice application of AI to do survey research but the lack of a systematic analysis makes me worry about the general validity of the results.

**Questions:**

.

**Ai Review Score:**

0

**Ethical Concerns:**

.

**Limitations:**

.

**Quality:**

2

**Strengths And Weaknesses:**

.

---

### Official Review · Reviewer_AIRev1 · 2025-10-06
**AIRev 1**

**Confidence:** 5
**Overall:** 3
**Clarity:** 0
**Significance:** 0
**Originality:** 0

**Summary:**

Summary by AIRev 1

**Questions:**

N/A

**Ai Review Score:**

3

**Quality:**

0

**Strengths And Weaknesses:**

This paper presents a structured literature synthesis of 124 knowledge tracing (KT) models/variants, introducing an eight-dimension context taxonomy and harmonizing heterogeneous results via normalized ranks and context-level, quality-weighted win rates. The main findings are context-specific: attention/Transformers excel on large/long logs, graph-based KT with reliable structure, time-aware/forgetting models under irregular gaps, content/LLM-augmented KT on text/code/dialogue, mixture-of-experts for heterogeneous cohorts, and logistic/factorization models in data-constrained settings. The paper is praised for its timely framing, clear taxonomy, methodological care, breadth, and transparency about limitations. However, it is criticized for lacking quantitative aggregate reporting (no per-context tables/figures or uncertainty), ad hoc weighting/capping choices, risk of bias from within-paper comparisons, unvalidated data extraction/coding, metric heterogeneity, incomplete corpus coverage, incremental insights, and minor reference/operational issues. The reproducibility claim is noted but artifacts are not yet accessible. Recommendations include reporting quantitative aggregates, providing corpus/accountability details, validating extraction/coding, addressing bias quantitatively, adding case studies, and cleaning up references/artifacts. Overall, the synthesis is promising but currently too qualitative and lacking in quantitative rigor and reliability checks to meet the standards of a selective venue.

---

### Official Review · Reviewer_AIRev2 · 2025-10-06
**AIRev 2**

**Confidence:** 5
**Overall:** 6
**Clarity:** 0
**Significance:** 0
**Originality:** 0

**Summary:**

Summary by AIRev 2

**Questions:**

N/A

**Ai Review Score:**

6

**Quality:**

0

**Strengths And Weaknesses:**

This paper presents a large-scale, structured synthesis of the knowledge tracing (KT) literature, analyzing 124 papers to determine which KT models perform best in various contexts. The authors categorize papers and datasets along eight contextual dimensions and use a rigorous two-stage aggregation methodology, including a quality-weighted scheme that rewards robust evaluation protocols. Key findings are that no single model family is universally superior; performance is highly context-dependent. Attention/Transformer models excel on large-scale, long-sequence data; graph-based models perform best with reliable structural information; and simpler factorization models remain competitive in data-constrained or interpretability-critical settings. The paper is significant for its methodological rigor, clarity, originality, and commitment to reproducibility. Minor weaknesses include the unavoidable potential for publication bias and the necessary aggregation of models into broad families, which may obscure some nuances. Overall, this is a landmark, technically sound, and highly impactful paper, exemplary in its execution and presentation, and is highly recommended.

---

### Official Review · Reviewer_AIRev3 · 2025-10-06
**AIRev 3**

**Confidence:** 5
**Overall:** 4
**Clarity:** 0
**Significance:** 0
**Originality:** 0

**Summary:**

Summary by AIRev 3

**Questions:**

N/A

**Ai Review Score:**

4

**Quality:**

0

**Strengths And Weaknesses:**

This paper presents a systematic literature synthesis of 124 knowledge tracing (KT) models to understand which models work best in different contexts. The work is conducted primarily by an LLM with human-LLM partnership and aims to provide practical guidance for context-aware KT model selection.

Quality: The paper is technically sound with a well-structured methodology. The approach of synthesizing literature rather than re-implementing models is reasonable given the practical constraints. The authors develop a comprehensive 8-dimensional context taxonomy and implement quality-aware weighting schemes to address evaluation biases. The methodology for harmonizing heterogeneous metrics and ranking systems is appropriate. However, some methodological choices appear arbitrary (as the authors acknowledge), particularly the novel weighting schemes that lack empirical validation.

Clarity: The paper is generally well-organized and clearly written. The context taxonomy is well-presented in Table 1, and the methodology is sufficiently detailed for understanding. The results section effectively summarizes findings for different contexts. However, some technical details about the aggregation methodology could be clearer, and the heavy reliance on AI-generated content occasionally results in surface-level explanations that require more depth.

Significance: This work addresses an important practical problem in educational technology. The synthesis of 124 KT models spanning three decades provides valuable insights for practitioners selecting appropriate models. The context-dependent findings (e.g., attention models for large-scale data, graph models for structured domains) offer actionable guidance. The identification of evaluation pitfalls and the quality-aware weighting approach contribute methodologically to the field.

Originality: While systematic reviews exist in this domain, the comprehensive scope (124 models), novel context taxonomy, and AI-powered synthesis approach provide original contributions. The quality-aware aggregation methodology and context-dependent analysis framework are innovative, though some elements lack theoretical grounding.

Reproducibility: The authors provide detailed methodology and promise to release data, code, and supplementary materials. The systematic approach with explicit inclusion/exclusion criteria supports reproducibility, though the stochastic nature of LLM involvement may introduce some variability in replication.

Ethics and Limitations: The authors appropriately discuss limitations including potential biases from metric heterogeneity, exclusion of papers without baselines, and focus on predictive performance over other important factors (fairness, interpretability, computational costs). The AI involvement is transparently disclosed through the checklist.

Citations and Related Work: The paper appropriately cites relevant prior work and positions itself well within the literature. The comprehensive coverage of 124 models demonstrates thorough literature coverage.

Concerns:
1. Some methodological choices (particularly weighting schemes) appear arbitrary and lack validation
2. Heavy AI involvement raises questions about depth of analysis and potential biases
3. The focus on predictive performance may not fully capture practical deployment considerations
4. The synthesis approach, while practical, cannot fully replace controlled empirical comparisons

Strengths:
1. Addresses an important practical problem with comprehensive scope
2. Novel methodology combining systematic review with AI assistance
3. Well-structured context taxonomy providing actionable insights
4. Transparent about limitations and AI involvement
5. Quality-aware approach to address evaluation biases

The paper makes a solid contribution to the knowledge tracing field by providing practical guidance for model selection across different contexts. Despite some methodological concerns, the comprehensive scope and novel approach provide value to the community.

---

### Note · Reviewer_AIRevCorrectness · 2025-10-06

**Correctness Check**

### Key Issues Identified:

- Arbitrary, unvalidated quality weights (e.g., ×1.25/×0.50; ×1.10/×0.75) with no empirical calibration; sensitivity results are asserted but not quantitatively reported.
- Capping rule (k=3) per (dataset × family × context) may introduce selection bias and causes families to be evaluated on different subsets of instances within the same context, complicating comparability.
- Insufficient detail on literature search reproducibility (e.g., exact queries, databases per query, inter-rater agreement) and on human/AI validation of extracted fields; heavy reliance on AI extraction without quantified error rates.
- Weighted median of ranks is not fully specified (algorithmic details; weight normalization), and notation inconsistencies (ai vs αi).
- No tables/figures reporting the actual context-level weighted win rates and weighted median ranks; sensitivity analyses are described but numerical outcomes are not provided.
- Publication bias not formally addressed (e.g., no funnel/selection diagnostics); reliance on within-paper comparisons only may under-represent newer families.
- Context label imputation is described as conservative, but no dedicated sensitivity analysis for these imputations is reported.
- Minor bibliographic irregularities/duplicates; while not critical, they slightly detract from formal polish and traceability.

---

### Note · Reviewer_AIRevRelatedWork · 2025-10-06

**Related Work Check**

No hallucinated references detected.

---

### Decision · Program_Chairs · 2025-10-08

**Decision:**

Reject

**Comment:**

Thank you for submitting to Agents4Science 2025! We regret to inform you that your submission has not been accepted. Please see the reviews below for more information.